# Contributions of Emotional Overload, Emotion Dysregulation, and Impulsivity to Eating Patterns in Obese Patients with Binge Eating Disorder and Seeking Bariatric Surgery

**DOI:** 10.3390/nu12103099

**Published:** 2020-10-12

**Authors:** Farid Benzerouk, Zoubir Djerada, Eric Bertin, Sarah Barrière, Fabien Gierski, Arthur Kaladjian

**Affiliations:** 1Psychiatry Department, Reims University Hospital, EPSM Marne, 51100 Reims, France; sbarriere@chu-reims.fr (S.B.); fabien.gierski@univ-reims.fr (F.G.); kaladjiana@epsm-marne.fr (A.K.); 2Cognition Health and Society Laboratory (EA 6291), Université de Reims Champagne-Ardenne, 51100 Reims, France; 3Department of Pharmacology E.A.3801, SFR CAP-santé, Reims University Hospital, 51100 Reims, France; zdjerada@chu-reims.fr; 4Champagne Ardenne Specialized Center in Obesity, University Hospital Center, 51100 Reims, France; ebertin@chu-reims.fr; 5INSERM U1247 GRAP, Research Group on Alcohol and Pharmacodependences, Université de Picardie Jules Verne, 80000 Amiens, France

**Keywords:** obesity, bariatric surgery, binge eating disorder, emotion dysregulation, emotional eating, external eating, strategies, non-acceptance, impulsivity

## Abstract

Background: Binge eating disorder (BED) is very frequently observed in patients considered for weight loss surgery and seems to influence their outcome critically. Literature highlights a global emotional overload in individuals with BED, but little is known on the mechanisms involved. The present study aimed to focus on emotion regulation, impulsivity, depression, and anxiety in people with and without BED and fulfilling inclusion criteria for bariatric surgery. Doing so, we sought to individualize factors related to BED. Then, we examined the contribution of depression, anxiety, emotion regulation difficulties, and impulsivity to inappropriate eating behaviors observed in patients with BED. Methods: A sample of 121 individuals (79.3% female, mean age: 40.82 ± 9.26, mean current body mass index (BMI): 44.92 kg/m^2^ ± 7.55) seeking bariatric surgery were recruited at the Champagne Ardenne Specialized Center in Obesity in Reims, France from November 2017 to October 2018. They were stratified as with or without BED according to the binge eating scale. Characteristics identified in univariate analyses as differentiating the two groups were then included in multivariable analyses. Results: Multivariable analyses showed that limited access to emotional regulation strategies was significantly associated with BED. Furthermore, inappropriate eating behaviors were independently associated with age, depression severity, anxiety, emotional dysregulation, and impulsivity in BED group. Conclusions: The present findings are indicative of an association between emotion deficit and BED in obese patients seeking bariatric surgery. Patients with BED could benefit from the addition of an emotion regulation intervention.

## 1. Introduction

Binge eating disorder (BED) involves frequent overeating during a discreet period of time (at least once a week for three months), combined with a lack of control, and is associated with three or more of the following items: eating more rapidly than normal; eating until feeling uncomfortably full; eating large amounts of food when not feeling physically hungry; eating alone because of feeling embarrassed by how much one is eating; and feeling disgusted with oneself, depressed, or very guilty afterward [1]. BED also causes significant distress [1] and is associated with various inappropriate eating behaviors. It is more common in females (3.5%) than in males (2.0%) and in obese individuals (5% to 30%) [2,3], especially those who are severely obese and those seeking obesity treatment: 17% at the time of surgery [4,5]. Moreover, BED seems to influence success after weight loss surgery [6]. Accordingly, it should be of interest to assess why people suffering from this disorder engage in various inappropriate eating behaviors in order to find a way to help them give up these harmful behaviors.

Inappropriate eating behaviors are like engaging in emotional eating or in binge eating. As a matter of fact, emotional eating behavior, the tendency to overeat in response to negative emotions, appears to be common in bariatric candidates (see for review [7]). Moreover, obese people with BED who are candidates for bariatric surgery are more likely to have severe binge eating symptoms than obese non-surgical individuals [7]. Bariatric surgery candidates also have more objective and subjective binge eating episodes per month than non-surgical weight loss patients [8].

According to the Diagnostic and Statistical Manual of Mental Disorders, DSM-5 criteria, binge eating and negative emotions are interconnected [1]. In their review, Dingemans et al. (2017) pointed that (1) several authors, using experimental studies, emphasized a relationship between emotional factors and overeating in individuals with BED; (2) these individuals were characterized by a higher prevalence of psychiatric comorbidities, exhibited higher levels of depression and anxiety; (3) they reported poorer mood especially prior to binge eating and can experience more negative stressors than subjects without BED; and (4) they can also feel more negative emotions (i.e., anger and/or frustration) related to interpersonal experiences [9]. Taken together, these results highlight a global emotional overload in individuals with BED. This emotional overload might increase the occurrence of binge eating. According to Polivy and Herman’s (1993) affect regulation model of binge eating, this behavior could be implemented to decrease emotional distress or negative affects [10].

Rather than focusing on emotions themselves in individuals with BED, other works have focused on emotion regulation. According to Gross [11], emotion regulation refers to “shaping which emotion one has, when one has them, and how one experiences or expresses these emotions”. Emotion regulation is conceptualized as involving emotion regulation abilities (i.e., the awareness and understanding of emotions, the acceptance of emotions, the ability to control impulsive behaviors and behave in accordance with desired goals when experiencing negative emotions), and emotion regulation strategies (i.e., use situationally appropriate emotion regulation strategies flexibly to modulate emotional responses as desired in order to meet individual goals and situational demands). Strategies can include adaptive ones such as reappraisal, problem-solving, and acceptance and maladaptive ones such as avoidance, rumination, and suppression. The relative absence of any or all of these abilities and strategies would indicate the presence of difficulties in emotion regulation, or an emotion dysregulation [12]. Regarding eating disorders, a recent meta-analysis by Prefit et al. [13] identified a transdiagnostic character of emotion regulation problems. Furthermore, compared to a control group without obesity, people suffering from obesity use significantly fewer cognitive emotion regulation strategies considered as adaptive regardless of their body mass index (BMI) [14] and they report using more emotional suppression [15]. Data also revealed that only emotional dysregulation significantly predicted binge eating vulnerability in a study involving 63 obese patients seeking surgical treatment [16]. Moreover, many studies identified lack of skills and strategies required to regulate negative affect adaptively and effectively (i.e., poorer emotional awareness and clarity, nonacceptance, difficulties with reappraisal, and with problem-solving) as being associated with eating disorders. Accordingly, individuals with disordered eating may have a greater vulnerability to using maladaptive emotion regulation strategies (i.e., rumination, avoidance of emotions, and suppression) [13].

In sum, the global emotional overload and dysfunctional emotion regulation abilities and strategies are then increasingly thought to be co-occurring risk factors in the onset and maintenance of BED by promoting maladaptive behaviors such as overeating and binge eating. However, according to Dingemans et al. [9], studies investigating the use of emotion regulation strategies amongst individuals with BED have found mixed results. Moreover, beyond emotional regulation, other researchers have outlined the role of impulsivity in this disorder. According to Giel et al. (2017), BED is also considered as a distinct phenotype, within the obesity spectrum, characterized by increased impulsivity and by an increased rash–spontaneous behavior in general and specifically toward food [17]. However, the simultaneous consideration of emotion regulation and impulsivity remains to be deepened. Therefore, there is a major interest to understand the impact of both emotion regulation as well as impulsivity in patients suffering from BED and seeking bariatric surgery.

Altogether, emotional overload (i.e., depression, anxiety), emotion regulation, and impulsivity may be associated with BED and could predispose individuals to developing and/or maintaining inappropriate eating behaviors among patients suffering from BED who are candidates for bariatric surgery. Given this, the primary aim of our study was to examine the associations with BED of emotional overload (depression, anxiety), emotion regulation, and impulsivity in obese people with and without BED. We expected people suffering from obesity with BED to present more depression, more anxiety, more emotion regulation difficulties, and more impulsivity than people suffering from obesity without BED (wBED). The second aim of our study was to examine the contribution of depression, anxiety, emotion regulation difficulties, and impulsivity to eating patterns observed in patients with BED. In this population, we sought to individuate which factors were significantly related to the assessed eating behaviors. More precisely, we expected that high levels of depression, anxiety, emotion regulation difficulties, and impulsivity were significantly related to emotional eating, external eating, and bulimic symptomatology. Improving our knowledge on BED is a necessary step to then develop indications of specific therapeutic strategies before surgery and, thus, allow their access to bariatric surgery and improve their outcomes after bariatric surgery.

## 2. Materials and Methods

### 2.1. Participants and Procedure

A sample of 121 obese candidates for bariatric surgery was recruited at the Champagne Ardenne Specialized Center in Obesity in Reims, France from November 2017 to October 2018. Participants were 79.3% female (*n* = 96), and ranged in age from 19 to 58, with a mean age of 40.82 (*SD* = 9.26). Mean BMI was 44.92 kg/m^2^ (*SD* = 7.55; range: 35.63–75.72).

The inclusion criteria were to be a candidate for bariatric surgery: obesity grade 2 (BMI 35.0–39.9 kg/m^2^) and at least one obesity-related comorbidity (e.g., hypertension, type 2 diabetes mellitus, dyslipidemia, sleep apnea), or obesity grade 3 (BMI ≥ 40.0 kg/m^2^). Obese patients had to be French-speaking females or males between the ages of 18 and 60. Participants with present or past drug or alcohol abuse or dependence were not included (as assessed by the Mini International Neuropsychiatric Interview (MINI) [18,19]), as these conditions would have been able to alter the assessments of emotion regulation and impulsivity, as well as participants having a bariatric procedure previously or a severe comorbid disorder such as neurologic impairments. The BMI was calculated by dividing the weight in kilograms by the square of the height in meters (BMI = weight (kg)/height (m^2^)) [20].

All study procedures were reviewed and approved by the local Institutional Review Board (IRB 2016-12). The study was carried out according to the Helsinki Declaration [21] and every patient included into the study provided written informed consent.

### 2.2. Measures

#### 2.2.1. Eating Behaviors

The Binge Eating Scale (BES; [22]) is a 16-item self-administered questionnaire used to assess the presence of binge eating behavior indicative of an eating disorder. Eight items describe behavioral manifestations (for example, eating fast or consuming large amounts of food) and eight items on associated feelings and cognitions (for example, fear of not stopping eating). Each item has a response range from 0 to 3 points (0 = no severity of the BES symptoms, 3 = serious problems on the BES symptoms). Marcus et al. (1988) created a range of scores for the BES from 0 to 46 points: a score of less than 17 points indicates minimal binge eating (BE) problems; a score between 18 and 26 points indicates moderate BE problems, and a score of more than 27 points indicates severe BE problems [23]. We considered binge eating as a categorical variable (significant binge eating if BES score ≥18). We used the validated French version [24]. The Cronbach’s alpha for the current study was 0.83.

The Bulimic Investigatory Test, Edinburgh (BITE; [25]) is a self-report questionnaire used to evaluate the presence and severity of bulimic symptomatology. It is composed of 33 items divided into two different subscales: a symptom subscale (30 items) and a severity subscale (3 items). Henderson and Freeman (1987) considered a BITE score under 10 points as indicative of no problem with eating behavior, a score between 10 and 20 points as indicative of abnormal eating patterns (from 15 to 20 points warns us of the presence of a possible subthreshold bulimia nervosa), and a score higher than 20 points constitutes altered eating patterns with a possible bulimia nervosa. The Cronbach’s alpha for the current study was 0.80.

The Dutch Eating Behavior Questionnaire (DEBQ) was administered using the French version to assess three components of eating behavior: emotional, external, and restrained eating [26,27]. It is a self-report measure that contains 33 items. Thirteen items assess emotional eating, 10 items assess external eating, and 10 items assess restrained eating. Each item is rated on a 5-point Likert scale. In the current study, Cronbach’s alpha was 0.95 for emotional eating, 0.86 for external eating, and 0.86 for restrained eating.

#### 2.2.2. Emotion Regulation and Impulsivity

The Difficulty in Emotion Regulation Scale (DERS; [12]) is a 36-item self-report questionnaire. It assesses six different aspects of emotional regulation including non-acceptance of emotional responses (Non-Acceptance), difficulty engaging in goal-directed behavior (Goals), impulse control difficulties (Impulse), lack of emotional awareness (Awareness), limited access to emotional regulation strategies (Strategies), and lack of emotional clarity (Clarity). This scale has demonstrated good internal consistency, construct and predictive validity, and test–retest reliability. We used the validated French version [28]. The total score demonstrated a Cronbach’s alpha of 0.92 in the present study. It was 0.90, 0.78, 0.84, 0.66, 0.86, and 0.66 respectively for the six different subscales described above.

The UPPS-P Impulsive Behavior Scale [29] consists of 20 items that evaluate different facets of impulsivity labeled Negative Urgency (4 items), Positive Urgency (4 items), (lack of) Premeditation (4 items), (lack of) Perseverance (4 items), and Sensation seeking (4 items). Cronbach’s alphas in our sample were 0.83, 0.74, 0.77, 0.79, 0.77, respectively.

### 2.3. Other Assessments (Depression Severity and Anxiety Levels)

Depression severity was assessed with the shortened Beck Depression Inventory (BDI). This is a widely used self-report scale consisting of 13 items [30], which has been validated in French [31]. The total score is obtained by adding the scores of the 13 items and ranges from 0 to 39, with higher scores indicating greater depression symptoms. The Cronbach’s alpha for the current study was 0.83.

Anxiety severity was assessed with the Spielberger State–Trait Inventory (STAI; [32]) which is a 40-item scale, using a 4-point Likert scale for each item. This scale was used to measure both trait anxiety (how dispositionally anxious a person is across time and situations) and state anxiety (how anxious a person is feeling at a particular moment). The Cronbach’s alpha for the current study was 0.93 for the trait anxiety scale and 0.73 for the state anxiety scale.

### 2.4. Statistical Analysis

Statistical analyses were performed with R 3.1.4 (The R Foundation for Statistical Computing, Vienna, Austria, http://www.r-project.org). Data summaries were presented as the mean and standard deviation (sd) for continuous measurements and as the frequency (percentage) for categorical variables. Categorical variables were analyzed using overall chi-squared (*χ*^2^).

Cronbach’s alpha was calculated for each scale and subscale (package psych).

Univariate analyses were done for continuous variables using the non-parametric Mann–Whitney U test. To avoid computational issues (model convergence failure due to sparse data), only covariates with at least five cases were considered in the model. Univariate analysis was performed to screen potential variables for inclusion in the final multivariable model.

Multivariable analyses were performed using logistic regression modeling (BED and wBED categories as the dependent variables), and the association between the identified variables and eating patterns in the BED group was assessed by multiple linear regression analysis while controlling the other covariates for confounding effects. Adjusted *β* coefficients (*β*adj) were estimated for all significant associations [33]. Among the variable selection procedures, backward elimination is preferred as it starts with the assumed unbiased global model [34,35]. The potential prognostic factors were established, and a multivariable model was derived by backward selection according to Akaike’s Information Criterion. For sensitivity, all identified associated covariates in the different model were also determined using the appropriate high-dimensional procedure as random forest (package randomForest) and sparse partial least squares discriminant analysis (package mixomics) and Lasso (package glmnet). The goodness-of-fit and appropriateness of the logistic regression model were evaluated using the Nagelkerke R squared and Hosmer–Lemeshow values and by the correct overall percentage of prediction. Multicollinearity was checked for all analyses. Variables significant at *p* = 0.05 at final multivariable analysis were retained as independent predictive factors. The Wald test was used for hypothesis testing. The stability and robustness of the model were validated using the technique of “bootstrap” resampling.

All *p*-values were two-tailed, with statistical significance indicated by a value of *p* < 0.05.

## 3. Results

### 3.1. Group Comparisons of Study Variables in Obese Adults Seeking Bariatric Surgery (with Binge Eating Disorder (BED) and without BED (wBED))

The stratification procedure based on the BES score (cutoff = 18) led to the identification of a group of 27 participants with BED and a group of 94 participants without BED (wBED) and to a prevalence of BED of 22.31%. There were no significant differences between BED and wBED groups with regard to current body mass index (kg/m^2^; mean ± SD) (45.83 ± 8.21 in the BED group and 44.66 ± 7.38 in the wBED group, *p* = 0.513) and maximum body mass index (kg/m^2^; mean ± SD) (48.19 ± 9.65 in the BED group and 46.87 ± 7.74 in the wBED group, *p* = 0.806). The comparison of demographic and clinical characteristics of the two groups is presented in Table 1.

In the univariate analysis, patients with BED showed significantly higher levels of depression, anxiety, eating-disorder symptoms (i.e., Emotional eating, External eating, BITE Symptom score, and BITE Severity score), emotion dysregulation, and impulsivity than patients without BED. In the multivariable analysis, the BITE symptom subscale and limited access to emotional regulation Strategies were significantly associated with BED with adjusted odds ratios (OR) of 1.517 (1.241–1.990), *p* < 0.001, and 1.176 (1.023–1.389), *p* = 0.03, respectively (Table 1 and Table 2).

### 3.2. Associations between Emotional Overload, Emotion Regulation Difficulties, Impulsivity, and Eating Patterns in Patients with BED

Univariate analyses concerning Emotional eating and External eating are shown in Table 3 and Table 4. Multivariable analysis showed that Emotional eating was independently associated with age (*β*adj = −0.415 (−0.792; −0.0390), DERS Non-Acceptance subscale score (*β*adj = 0.883 (0.060–1.705)), and UPPS-P Lack of premeditation subscale score (*β*adj = 3.750 (1.838–5.661)) (Table 3). Multivariable analysis showed that External eating was independently associated with the anxiety trait (*β*adj = 0.319 (0.090–0.547)), DERS Impulse control difficulties subscale score (*β*adj = 0.431 (0.115–0.746)), and UPPS-P Negative Urgency subscale score (*β*adj = 0.996 (0.496–1.497)) (Table 4).

Univariate analyses concerning BITE symptoms and BITE severity are shown in Table 5 and Table 6. Multivariable analysis showed that BITE symptom score was independently associated with depression severity score (*β*adj = 0.255 (0.082–0.428)), DERS Non-Acceptance subscale score (*β*adj = 0.299 (0.131–0.466)), and UPPS-P Lack of premeditation subscale score (*β*adj = 1.469 (0.851–2.087)) (Table 5). Multivariable analysis showed that BITE Severity score was independently associated with the trait anxiety (*β*adj = 0.176 (0.061–0.290)), DERS Impulse control difficulties subscale score (*β*adj = 0.587 (0.415–0.759)), DERS Clarity subscale score (*β*adj = 0.304 (0.094–0.514)), and UPPS Negative Urgency subscale score (βadj = −0.832 (−1.083; −0.582)) (Table 6).

## 4. Discussion

The primary aim of the present study was to examine the contributions of emotional overload (depression and anxiety), emotion regulation, and impulsivity in female and male obese people with and without BED and seeking bariatric surgery. Moreover, this study aimed to examine the contribution of emotional overload (depression and anxiety), emotion regulation difficulties, and impulsivity to eating patterns observed in patients with BED. Our two main findings, discussed below, were as follows: (1) limited access to emotional regulation strategies and bulimic symptoms were significant predictors of BED; (2) emotional eating, external eating, the degree of binge eating symptoms and the severity of bingeing and purging behaviors in patients with BED were associated with specific dimensions of emotion regulation and impulsivity as well as anxiety and depression scores. To the best of our knowledge, our study is the first to assess these contributions in this population.

More anecdotal, in our sample, the prevalence of BED was of 22.31%, which is consistent with the prevalence reported by the literature [36,37].

### 4.1. Emotional Overload, Emotion Regulation, and Impulsivity

The findings showed that emotion dysregulation (i.e., limited access to emotional regulation strategies) was a significant predictor of BED. The DERS Strategies subscale reflects limited access to the flexible use of adaptive emotion regulation skills to modulate (vs. eliminate) the intensity and/or temporal features of emotional responses [12]. This finding is consistent with results reported in a review from Dingemans et al. (2017) [9]. These authors suggest that individuals with BED are more likely to engage in maladaptive emotional strategies (e.g., suppression, rumination) and less likely to engage in adaptive ones (e.g., acceptance, reappraisal). Moreover, in a prospective study, Svaldi et al. (2019) demonstrated that, in individuals with BED, rumination was a significant predictor of binge eating and that from a clinical perspective, ruminations were correlated with the probability of a binge episode by approximately 28% [38]. In the specific population of patients seeking bariatric surgery, Cella et al. (2019) reported that patients suffering from BED exhibited more emotional dysregulation, as assessed by the Eating Disorders Inventory-3 (EDI-3), than patients without BED [16]. Moreover, Gianini et al. (2013) reported that, in treatment-seeking obese adults with BED, limited access to emotion regulation strategies was strongly associated with emotional overeating [39].

When examining eating-related behaviors, as assessed by the BITE, only bulimic symptoms were associated with BED, which is not surprising as binge eating is the essential feature of this disorder [1]. The severity subscale score was not associated with BED. It outlines that individuals with BED, in our sample, as was logically expected, do not show marked or sustained dietary restriction designed to influence body weight and shape between binge eating episodes.

In our univariate analysis, patients with BED showed significantly higher levels of depression, anxiety, and impulsivity than patients without BED. We expected that these dimensions would also be significant predictors to BED in the multivariable analysis, but this was not the case. This result underlines, in our population, the major contribution of the emotional regulation dimension to the disorder.

### 4.2. Emotional Overload, Emotion Regulation Difficulties, Impulsivity, and Eating Patterns in Patients with BED

The second aim of our study was to examine the contribution of emotional overload, emotion regulation difficulties, and impulsivity to eating patterns observed in patients with BED. The evaluated eating patterns were emotional eating and external eating as assessed by the DEBQ [26,27] and the degree of binge eating symptoms and the severity of bingeing and purging behaviors (as defined by their frequency) as assessed by the BITE [25].

Emotional eating was independently associated with age, DERS Non-acceptance subscale score, and UPPS-P Lack of premeditation subscale score. In our study, younger-old adults exhibited more emotional eating compared to older-old ones. This tendency to overeat in response to negative emotions appears to be more frequent in young adults than in older adults and it may be due to an increase in the use of emotion regulation skills with age [40]. The DERS Non-acceptance subscale is related to coping style [11] (p. 337). Coping is generally viewed as an individual’s effort (cognitively and/or behaviorally) to adapt to or reduce distress in response to stressful events [41]. Hence, a maladaptive coping style to emotions in patients with BED could trigger an emotional eating pattern. However, in the present study, we did not have the information about what stressful events or negative thoughts patients needed to deal with and what coping styles they usually used. Another dimension contributing to emotional eating was lack of premeditation. This dimension is defined as the tendency to act without thinking and is viewed as presenting deficits in conscientiousness [42]. These deficits could lead to decision-making with little regard to past outcomes or forethought for possible future outcomes. It could also reflect a high tolerance for punishment from maladaptive behaviors (i.e., the negative consequences of these behaviors may not be sufficient to deter individuals with high scores on this dimension) [43].

External eating, corresponding to overeating in response to food-related cues such as the sight and smell of attractive food, was independently associated with the anxiety trait, DERS Impulse control difficulties subscale score, and UPPS-P Negative urgency subscale score. Interestingly, Heeren et al. (2018) found that the trait anxiety can be conceptualized as a single and coherent network system of interacting elements [44]. Noteworthily, they reported that the presence of intrusive thoughts and being unable to get disappointments out of one’s mind emerged as the most central features of the trait anxiety network. It could mean that craving induced by food cues and negative affectivity may predispose to this eating style. These features could be linked to the “food addiction” hypothesis [45]. The difficulties maintaining behavioral control when distressed, assessed by the DERS Impulse control difficulties subscale, describe individuals who have very strong feelings that are hard to control [11] (p. 337). Moreover, this subscale specifically focuses on feeling “out of control” in emotionally distressing situations. In our sample, it is another dimension that predisposes patients with BED to eat in response to food-related cues. Negative urgency refers to acting rashly and impulsively when in extreme distress and involves impaired inhibitory control [46]. Patients with BED seem to use palatable food to compensate for negative affect or use food in a comforting fashion to cope with life distress. Taken together, negative affect and cravings induced by food cues increase the likelihood of external eating.

The degree of binge eating symptoms was independently associated with depression score, DERS Non-acceptance subscale score and UPPS-P Lack of premeditation subscale score. Interestingly, two dimensions (i.e., DERS Non-acceptance subscale score and UPPS-P Lack of premeditation subscale score) are the same that for emotional eating. Decision-making with little regard for past outcomes or forethought for possible future outcomes and a high tolerance for punishment from maladaptive behaviors seem then to also contribute to the likelihood of having a severe binge eating behavior. These results contribute to enriching our understanding of this eating behavior. Moreover, they raise the question of the links between emotional eating and binge eating and of the underlying psychopathology of these eating patterns. Depression severity was also associated with the binge eating behavior as expected based on literature data [22,47].

Finally, the severity of bingeing and purging behaviors was independently associated with the trait anxiety, DERS Impulse control difficulties subscale score, DERS Clarity subscale score, and UPPS-P Negative urgency subscale score. Three dimensions are common with those identified as being associated with External eating (i.e., anxiety trait, DERS Impulse subscale score, and UPPS Negative urgency subscale score). These dimensions are then risk factors for developing a highly disordered eating pattern and a presence of binge eating. However, unexpectedly, there was a reversed link with negative urgency. This may be due to the behaviors assessed by this score. In fact, it provides an index of the severity as defined by the frequency of binge eating and purging behavior. Among the purging behaviors, is the use of fasting. Therefore, we could make the assumption that patients with BED and with low levels of negative urgency may be more prone to using fasting to control their weight. It could be a reason why our two populations of patients (BED vs. wBED) did not differ in current and past BMI. The emotional clarity subscale predicted the severity of bingeing and purging behaviors. This dimension was strongly associated with emotional overeating [39] and could be, in our patients with BED, a risk factor of a high frequency of binge eating. However, this result is to be taken with caution, given the low Cronbach’s alpha observed for this dimension in our population.

### 4.3. Limitations

This study has several limitations. The main limitation is about the power. If we use the number of cases to estimate the a priori power, we estimate that we can study between 2 to 3 variables (epv = 27/9 = 3; 27/10 = 2.7). Calculating the a posteriori power (post hoc) using the IBM SPSS sample Power software, or using simulations, allows us to determine a power at 88% to capture the effect of the three most influential covariates in our multivariable model. Clearly, we lack the power to study the numerous covariates. To check the result of our main endpoint, we have therefore proposed methods suitable for multivariable analyses on databases with a lack of power, such as penalized regression methods like Lasso [48]. These sensitivity analyses confirm our results (Appendix A
Appendix A), but do not exclude, that the other covariates are not significant partly due to a lack of power. Further studies with more power will be needed to estimate the association of the other covariates with our main endpoint. The same results are also confirmed by the use of Bayesian statistical analysis performed by the BRMS package [49] (Appendix A
Appendix A) and, further, by selecting the covariates by bootstrapping using the rms package according to the methodology previously described [50]. The three most important parameters retained in the model are BITE symptom subscale (78.69%), DERS strategies (46.03%), and Emotional eating (34.92%). A second limitation is related to its cross-sectional nature; therefore, caution is needed in inferring causality. A third limitation is based on the assessment of BED, depression, anxiety, emotion regulation, impulsivity, and eating behavior styles through self-reports, which are subject to possible biases such as desirability or response bias. However, the validity of these questionnaires has been well supported in previous studies and our reliability indices were satisfactory except for DERS Lack of emotional clarity as mentioned earlier. Moreover, in future studies, these limitations could be overcome by using ecological momentary assessments considering patients’ natural environment. Such tools are, for example, validated in nutritional epidemiology and in psychiatry (e.g., depression) [51,52].

## 5. Conclusions

The results of the present study suggest that obese patients with BED who are seeking bariatric surgery are characterized by a limited global access to the flexible use of adaptive emotion regulation skills to modulate features of emotional responses. Moreover, in this population, many dimensions of emotion regulation are associated with different pathological eating patterns (with sometimes the same dimension associated with varying patterns of eating), which emphasizes the pleiotropic side of these dimensions. The same results are observed for the anxiety trait and impulsivity. Taken together, our results lead us to believe that patients with BED could benefit from the addition of an emotion regulation intervention, which could significantly improve their eating behaviors before surgery. It could also improve the outcomes of bariatric surgery [53]. Further research is needed to confirm our findings, to implement and evaluate emotion regulation interventions, and to characterize better neural correlates of emotion regulation in patients with BED.

## Figures and Tables

**Table 1 nutrients-12-03099-t001:** Comparison of sociodemographic and clinical characteristics in obese adults seeking bariatric surgery using univariate analyses (with binge eating disorder (BED) and without BED (wBED)).

Variables	BED (*n* = 27)	wBED (*n* = 94)	Univariate Analysis
			*p*
Age	43.19 ± 9.80	40.14 ± 9.04	0.095
Female/Male	18/9	78/16	0.066
Beck Depression Inventory	10.96 ± 6.22	5.47 ± 4.81	<0.001
State anxiety inventory (STAI-A)	42.33 ± 13.34	32.41 ± 9.16	0.001
Trait anxiety inventory (STAI-B)	47.74 ± 5.95	41.26 ± 9.27	0.002
BITE total score	18.07 ± 6.36	8.61 ± 4.81	<0.001
BITE Symptom subscale	14.67 ± 4.25	6.88 ± 3.97	<0.001
BITE Severity subscale	3.41 ± 3.75	1.72 ± 2.02	0.006
DEBQ total score	96.81 ± 16.13	78.91 ± 15.38	<0.001
Emotional eating	39.56 ± 12.50	26.30 ± 9.64	<0.001
External eating	28.33 ± 5.52	23.83 ± 6.18	0.001
Restrained eating	28.93 ± 5.42	28.79 ± 8.27	0.822
DERS			
Total score	93.11 ± 18.92	76.27 ± 16.26	<0.001
Non-acceptance	14.67 ± 6.48	10.32 ± 4.32	<0.001
Goals	13.93 ± 4.05	11.16 ± 3.82	0.003
Impulse	14.26 ± 4.60	10.96 ± 3.65	0.001
Awareness	18.07 ± 5.05	18.14 ± 4.49	0.854
Strategies	19.59 ± 5.79	14.54 ± 4.34	<0.001
Clarity	12.59 ± 3.52	11.15 ± 3.25	0.097
UPPS-P			
Total score	31.11 ± 6.18	27.10 ± 8.31	0.005
Negative Urgency	6.22 ± 2.95	4.65 ± 2.87	0.008
Positive Urgency	6.67 ± 2.13	5.69 ± 2.62	0.077
Lack of Premeditation	6.81 ± 1.75	6.16 ± 1.92	0.082
Lack of Perseverance	6.26 ± 2.03	5.69 ± 1.83	0.165
Sensation Seeking	5.15 ± 2.75	4.90 ± 2.64	0.718

Note. Data presented as mean ± sd for quantitative variables and percentages for qualitative variables. BITE: Bulimic Investigatory Test, Edinburgh; DEBQ: Dutch Eating Behavior Questionnaire; DERS: Difficulty in Emotion Regulation Scale; UPPS-P: UPPS Impulsive Behavior Scale.

**Table 2 nutrients-12-03099-t002:** Comparison of sociodemographic and clinical characteristics in obese adults seeking bariatric surgery using multivariable analyses (with binge eating disorder (BED) and without BED (wBED)).

Variables	BED (*n* = 27)	wBED (*n* = 94)	Multivariable Analysis	
			*p*	Adjusted OR
Age	43.19 ± 9.80	40.14 ± 9.04	0.314	
Female/Male	18/9	78/16	0.259	
Beck Depression Inventory	10.96 ± 6.22	5.47 ± 4.81	0.520	
State anxiety inventory (STAI-A)	42.33 ±13.34	32.41 ± 9.16	0.987	
Trait anxiety inventory (STAI-B)	47.74 ± 5.95	41.26 ± 9.27	0.439	
BITE total score	18.07 ± 6.36	8.61 ± 4.81		
BITE Symptom subscale	14.67 ± 4.25	6.88 ± 3.97	<0.001	1.517 (1.241–1.9900
BITE Severity subscale	3.41 ± 3.75	1.72 ± 2.02	0.719	
DEBQ total score	96.81 ±16.13	78.91 ± 15.38		
Emotional eating	39.56 ±12.50	26.30 ± 9.64	0.076	
External eating	28.33 ± 5.52	23.83 ± 6.18	0.880	
Restrained eating	28.93 ± 5.42	28.79 ± 8.27		
DERS				
Total score	93.11 ±18.92	76.27 ± 16.26		
Non-acceptance	14.67 ± 6.48	10.32 ± 4.32	0.206	
Goals	13.93 ± 4.05	11.16 ± 3.82	0.685	
Impulse	14.26 ± 4.60	10.96 ± 3.65	0.296	
Awareness	18.07 ± 5.05	18.14 ± 4.49		
Strategies	19.59 ± 5.79	14.54 ± 4.34	0.03	1.176 (1.023–1.389)
Clarity	12.59 ± 3.52	11.15 ± 3.25	0.662	
UPPS-P				
Total score	31.11 ± 6.18	27.10 ± 8.31		
Negative Urgency	6.22 ± 2.95	4.65 ± 2.87	0.661	
Positive Urgency	6.67 ± 2.13	5.69 ± 2.62	0.833	
Lack of Premeditation	6.81 ± 1.75	6.16 ± 1.92	0.650	
Lack of Perseverance	6.26 ± 2.03	5.69 ± 1.83	0.449	
Sensation Seeking	5.15 ± 2.75	4.90 ± 2.64		

Note. Data presented as mean ± sd for quantitative variables and percentages for qualitative variables. OR: odds ratio; BITE: Bulimic Investigatory Test, Edinburgh; DERS: Difficulty in Emotion Regulation Scale; UPPS-P: UPPS Impulsive Behavior Scale. In the final multivariable model using logistic regression modeling (BED and wBED categories as the dependent variables), potential covariates were age (years), sex (female: 2, male: 1), Beck Depression Inventory score, STAI-A score, STAI-B score, BITE subscales scores, Emotional eating score, External eating score, Non-acceptance score, Goals score, Impulse score, Strategies score, Clarity score, Negative Urgency score, Positive Urgency score, Lack of Premeditation score, and Lack of Perseverance score. Only significant adjusted ORs are presented.

**Table 3 nutrients-12-03099-t003:** Multiple linear regression analysis of the association between DEBQ subscale (Emotional eating) and sociodemographic and clinical characteristics in the BED group.

Covariates	Univariate Analysis	Multivariable Analysis
	*p*	*β*adj (95% CI), *p*
Age	0.043	−0.415 (−0.792; −0.039), *p* = 0.042
Female/Male	0.290	
Beck Depression Inventory	0.181	
State anxiety inventory (STAI-A)	0.472	
Trait anxiety inventory (STAI-B)	0.751	
DERS		
Non-Acceptance	0.067	0.883 (0.060–1.705), *p* = 0.048
Goals	0.648	
Impulse	0.067	
Awareness	0.571	
Strategies	0.810	
Clarity	0.529	
UPPS-P		
Negative Urgency	0.404	
Positive Urgency	0.881	
Lack of Premeditation	0.004	3.750 (1.838–5.661), *p* < 0.001
Lack of Perseverance	0.157	
Sensation Seeking	0.691	

Note. Data are presented as *p*-value in univariate analysis and in adjusted regression *β* coefficients (*β*adj), 95% confident interval (CI), and *p*-value in multivariable analysis (only significant *β* coefficients (*β*adj), 95% confident interval are presented). OR: Odds ratio; DERS: Difficulty in Emotion Regulation Scale; UPPS-P: UPPS Impulsive Behavior Scale. In the final multivariable model of Emotional eating, potential covariates were age (years), Beck Depression Inventory score, Non-acceptance score, Impulse score, Lack of Premeditation score and Lack of Perseverance score.

**Table 4 nutrients-12-03099-t004:** Multiple linear regression analysis of the association between DEBQ subscale (External eating) and sociodemographic and clinical characteristics in the BED group.

Covariates	Univariate Analysis	Multivariable Analysis
	*p*	*β*adj (95% CI), *p*
Age	0.366	
Female/Male	0.666	
Beck Depression Inventory	0.308	
State anxiety inventory (STAI-A)	0.608	
Trait anxiety inventory (STAI-B)	0.184	0.319 (0.090–0.547), *p* < 0.05
DERS		
Non-Acceptance	0.035	
Goals	0.557	
Impulse	0.003	0.431 (0.115–0.746), *p* < 0.05
Awareness	0.624	
Strategies	0.178	
Clarity	0.690	
UPPS-P		
Negative Urgency	<0.001	0.996 (0.496–1.497), *p* < 0.001
Positive Urgency	0.053	
Lack of Premeditation	0.175	
Lack of Perseverance	0.532	
Sensation Seeking	0.874	

Note. Data are presented as *p*-value in univariate analysis and in adjusted regression *β* coefficients (*β*adj), 95% confident interval, and *p*-value in multivariable analysis (only significant *β* coefficients (*β*adj), 95% confident interval are presented). OR: odds ratio; DERS: Difficulty in Emotion Regulation Scale; UPPS-P: UPPS Impulsive Behavior Scale. In the final multivariable model of External eating, potential covariates were STAI-B score, Non-acceptance score, Impulse score, Strategies score, Negative Urgency score, Positive Urgency score, and Lack of Premeditation score.

**Table 5 nutrients-12-03099-t005:** Multiple linear regression analysis of the association between BITE subscale (symptom score) and sociodemographic and clinical characteristics in the BED group.

Covariates	Univariate Analysis	Multivariable Analysis
	*p*	*β*adj (95% CI), *p*
Age	0.734	
Female/Male	0.347	
Beck Depression Inventory	0.085	0.255 (0.082–0.428), *p* < 0.01
State anxiety inventory (STAI-A)	0.015	
Trait anxiety inventory (STAI-B)	0.194	
DERS		
Non-Acceptance	0.068	0.299 (0.131–0.466), *p* < 0.01
Goals	0.745	
Impulse	0.085	
Awareness	0.774	
Strategies	0.279	
Clarity	0.160	
UPPS-P		
Negative Urgency	0.297	
Positive Urgency	0.72	
Lack of Premeditation	0.003	1.469 (0.851–2.087), *p* < 0.001
Lack of Perseverance	0.797	
Sensation Seeking	0.398	

Note. Data are presented as *p*-value in univariate analysis and in adjusted regression *β* coefficients (*β*adj), 95% confident interval, and *p*-value in multivariable analysis (only significant *β* coefficients (*β*adj), 95% confident interval are presented). OR: odds ratio; BITE: Bulimic Investigatory Test, Edinburgh; DERS: Difficulty in Emotion Regulation Scale; UPPS-P: UPPS Impulsive Behavior Scale. In the final multivariable model of BITE symptom subscale, potential covariates were Beck Depression Inventory score, STAI-A score, STAI-B score, Non-acceptance score, Impulse score, Clarity score, and Lack of Premeditation score.

**Table 6 nutrients-12-03099-t006:** Multiple linear regression analysis of the association between BITE subscale (severity score) and sociodemographic and clinical characteristics in the BED group.

Covariates	Univariate Analysis	Multivariable Analysis
	*p*	*β*adj (95% CI), *p*
Age	0.561	
Female/Male	0.698	
Beck Depression Inventory	0.064	
State anxiety inventory (STAI-A)	0.053	0.176 (0.061–0.290), *p* < 0.01
Trait anxiety inventory (STAI-B)	0.118	
DERS		
Non-Acceptance	0.146	
Goals	0.443	
Impulse	0.003	0.587 (0.415–0.759), *p* < 0.001
Awareness	0.052	
Strategies	0.089	
Clarity	0.034	0.304 (0.094–0.514), *p* <0.01
UPPS-P		
Negative Urgency	0.078	−0.832 (−1.083; −0.582), *p* < 0.001
Positive Urgency	0.93	
Lack of Premeditation	0.907	
Lack of Perseverance	0.806	
Sensation Seeking	0.547	

Note. Data are presented as *p*-value in univariate analysis and in adjusted regression *β* coefficients (*β*adj), 95% confident interval, and *p*-value in multivariable analysis (only significant *β* coefficients (*β*adj), 95% confident interval are presented). OR: odds ratio; BITE: Bulimic Investigatory Test, Edinburgh; DERS: Difficulty in Emotion Regulation Scale; UPPS-P: UPPS Impulsive Behavior Scale. In the final multivariable model of BITE severity subscale, potential covariates were Beck Depression Inventory score, STAI-A score, STAI-B score, Non-acceptance score, Impulse score, Awareness score, Strategies score, Clarity score, and Negative Urgency score.

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
