# Peer review of "Contributions of Emotional Overload, Emotion Dysregulation, and Impulsivity to Eating Patterns in Obese Patients with Binge Eating Disorder and Seeking Bariatric Surgery"

_nutrients, 2020, doi:10.3390/nu12103099_

Round 1
Reviewer 1 Report
As discussed in the previous round of revision, the limited sample size is a main limitation of this study, especially the reduced statistical power of multivariable models. For this reason, the Authors should assess the statistical power of their analyses and discuss this point as a main limitation.
Reviewer 2 Report
This is interesting study to investigate the contributions of emotional overload, emotion regulation and impulsivity in obese people with and without BED. This study carries out important meaning in clinical practice. The results are potentially interesting and straightforward. However, there are major concerns with respect to expression of tables.
It is so difficult to read and understand tables.
Table 1. Where is the p-value for comparison between BED and wBED?
There are two p-values. What each p-value means?
I can not understand to combine the comparison of clinical characteristics, univariate analysis, and multivariate analysis.
In table 2: Did the authors presented the only significant Beta_adj(95%CI)?. If so, what are the other p-values (eg. P=0.210, p-=0.942….)?
The expression of tables are so confusing and this reduced the merit of this study.
More clear expression of tables or detailed table legeds could help to increase the quality of this article.
Round 2
Reviewer 2 Report
Thank you for appropriate revision.
Author Response
See attached file

This manuscript is a resubmission of an earlier submission. The following is a list of the peer review reports and author responses from that submission.
Round 1
Reviewer 1 Report
This manuscript by Benzerouk and colleagues aims to investigate how emotional overload, emotion, dysregulation and impulsivity associate with eating patterns in obese patients with binge eating disorder and seeking bariatric surgery. In my opinion, the topic is generally interesting but I have the following concerns:
- The introduction is well written and describes the current state of the art using appropriate and relevant references. However, I would suggest to provide a more specific aim of the study.
- With respect to the method section, my crucial concern regards the sample size used to conduct the study. If a sample size of 120 participants could be sufficient to explore bivariate associations with a good statistical power, I think it is not adequate for multivariable analysis using a lot of independent variables. For this reason, I would suggest to estimate the adequate sample size a priori or to calculate the statistical power based on the current number of participants. I think that the limited sample size prevents that some relationships reach the significance level.
- Results are clearly described, however I found difficult to understand table in which the authors reported both univariate and mulativariable analyses. Please provide p-values for each variable included in the regression models instead of "NS"
- The discussion section describes the main findings in the context of previous research. Moreover, the Authors summarized the main limitations of their study (cross-sectional design and self-reported informations. Accordingly, I would suggest to introduce a brief discussion on some tools that could be useful to overcome these limitations. Please consider a recent review on the application of ecological momentary assessment in nutritional epidemiology (doi: 10.3390/nu11112696.)
Reviewer 2 Report
General Comments:
The article “Contributions of emotional overload, emotion dysregulation and impulsivity to eating patterns in obese patients with binge eating disorder and seeking bariatric surgery” by Benzerouk et al., in my opinion, is a well-written paper, which adds significant contributions to the topic.
The authors assessed eating behaviors, emotion regulation, impulsivity, and severity of depression and anxiety in obese patients with and without BED who sought for bariatric surgery. The results suggest that obese patients with BED may have difficulties regarding emotion regulation and adaptive skills which could be associated with pathological eating patterns. Also, higher levels of depression, anxiety, and impulsivity seem to be more common in this group when compared to obese patients without BED, but it is not a major predictor for BED.
Abstract:
The setting and time of the study must be included.
Introduction: well-described, well organized and it clearly describes all main concepts and background.
Materials and Methods: well-detailed and well-described.
A kind suggestion: consider placing paragraph in row 117 at the beginning of the results sections (There were no significant differences between BED and wBED …)
- I suggest adding brief examples regarding “obesity-related comorbidities” that were used as criteria.
Results:
245 please check; an “e” letter is missing in the word ‘eating”
Table 1. Please correct “age” to Age (with a capital letter)
In my digital format the columns of the tables are displaced leading to chopped words and numbers. Please, correct for the final version.
Please add NS and OR in footnotes.
Discussion:
282-283 It is not clear what does “even if we used a self-administered questionnaire” mean? You may consider elaborating a little bit more than 2 rows if the idea is to explain that your results were obtained using self-administered questionnaires.